# Efficacy and Feasibility of the Minimal Therapist-Guided Four-Week Online Audio-Based Mindfulness Program ‘Mindful Senses’ for Burnout and Stress Reduction in Medical Personnel: A Randomized Controlled Trial

**DOI:** 10.3390/healthcare10122532

**Published:** 2022-12-14

**Authors:** Pavinee Luangapichart, Nattha Saisavoey, Natee Viravan

**Affiliations:** Department of Psychiatry, Faculty of Medicine Siriraj Hospital, Mahidol University, Bangkoknoi, Bangkok 10700, Thailand

**Keywords:** online, mindfulness, burnout, stress, medical personnel

## Abstract

Previous online mindfulness-based interventions for burnout and stress reduction in medical personnel had limited effect size and high dropout rate, so we developed a new online mindfulness program ‘Mindful Senses (MS)’ with aims to increase effect size and lower dropout rate. To test its efficacy and feasibility, ninety medical personnel with moderate or high levels of burnout and stress from across Thailand were recruited and randomly allocated into Group A and Group B equally. Group A read psychological self-help articles (PSA) and attended MS program through smartphone application during weeks 1–4. Group B read PSA during weeks 1–4 and attended MS program during weeks 9–12. Burnout, stress, anxiety, depression, mindfulness, and quality of life were measured at baseline, week 4, and week 8 for both groups, and at weeks 12 and 16 for Group B. Group x time interaction was analyzed by repeated-measures ANOVA. The results showed that, compared to PSA only, MS + PSA had statistically significant improvement of burnout, stress, anxiety, depression, mindfulness, and quality of life with moderate-to-very large effect sizes at week 4 (d = 1.33, 1.42, 1.04, 1.14, 0.70, and 1.03, respectively) and moderate-to-large effect sizes at week 8 (d = 0.84, 0.98, 0.73, 0.73, 0.66, and 0.94, respectively). The dropout rate was 4.4%. In conclusion, the MS program has potential to be an alternative intervention for medical personnel suffering from burnout and stress.

## 1. Introduction

Burnout and stress are major psychological concerns in medical personnel. The percentage of physicians who suffer from burnout ranges from 0% to as high as 80.5% across the world [1]. In Southeast Asia, 6.4–12% of healthcare workers suffer from stress [2,3]. Burnout and stress in medical personnel do not impact only healthcare professionals, but also their patients and close ones who were affected by their decreased performance or function. Previous studies found many negative professional consequences due to burnout and stress, such as lower patient satisfaction, impaired quality of care, decreased professional work effort, and increased medical errors [4,5,6]. Additionally, burnout can lead to substance abuse, broken relationships, marital and family problems, and even suicide [4,7]. The causes of burnout and stress included heavy workloads, inadequate staffing, difficult work conditions, low career satisfaction, and interpersonal and professional conflicts [8]. The mental health issues of medical professionals became increasingly worrisome during the coronavirus disease 2019 (COVID-19) pandemic. Healthcare professionals exposed to patients being tested for COVID-19 had a 1.4–1.6 times higher prevalence of burnout and stress [5], and the proportion of medical personnel suffering from burnout and high stress increased to 37.4–75% [6,9], and 40–46.1% [6,10], respectively.

Mindfulness-based interventions (MBI) are proven effective for burnout and stress reduction in medical personnel [11,12]. Mindfulness is defined as the awareness that emerges through paying attention on purpose, in the present moment, and non-judgmentally to the moment by moment unfolding of experience [13,14]. It is a form of mental practice rooted in Buddhist traditions in Asia, approximately 2500 years ago, and has taken root in the Western hemisphere since the late 19th century. In 1979, Jon Kabat-Zinn developed the first MBI, a psychological intervention using mindfulness as a core concept called Mindfulness-Based Stress Reduction (MBSR) [13,14]. Since then, other MBIs have been developed, for example, Mindfulness-Based Cognitive Therapy (MBCT) and Mindfulness Self Compassion (MSC). These MBIs have proven to be efficacious in reducing depression, anxiety, stress, and burnout [15,16,17]. MBIs have been continually developed, and there are various intervention platforms for MBIs nowadays. In addition to traditional face-to-face mindfulness groups, people interested in mindfulness can access online mindfulness programs by using computers, laptops, or mobile phones, which include self-help mindfulness programs, online mindfulness groups, or even mindfulness applications [15].

Online MBIs and mindfulness applications are increasingly common [18,19] because they provide alternative treatments for people that face barriers to mental health care service, which include time constraints, lack of access to services, confidentiality or social stigma concerns, and high costs [18,19]. Social distancing requirements and heightened stress during the COVID-19 pandemic may have further increased the appeal of online psychological interventions [14].

Previous studies have reported uncertain evidence of full online MBIs on burnout [20,21,22] and stress [23,24] in medical personnel. Earlier studies were limited by lack of randomization [22,23,24], lack of a control group [22,23,24] or attention control group [21], underpowered sample size [20,21,23], and had high dropout rates [22,24]. The outcome effect sizes in many studies have been merely small to medium [23,24]. Some study interventions have required at least eight weeks of program attendance [21,23,24], which may be difficult for healthcare professionals with limited time.

Therefore, we developed a new online MBI program called Mindful Senses (MS) with an aim to be the online MBI program that can reduce burnout and stress in medical personnel with larger effect size and lower dropout rate and tested its efficacy and feasibility in a randomized controlled trial (RCT). Additionally, previous studies [25,26] found that the effects of psychoeducation on stress reduction were equivocal, and there has been no study on the effects of psychoeducation alone on burnout. Hence, we explored whether reading Psychological Self-Help Articles (PSA) alone can reduce burnout and stress, and whether concurrently reading PSA and attending the MS program (MS + PSA) would boost the intervention effect over non-concurrently providing both interventions. We hypothesized that MS and PSA would reduce burnout and stress in medical personnel, and reading PSA along with attending the MS program would boost the intervention effects over non-concurrently providing MS and PSA.

## 2. Materials and Methods

This open-label, parallel-group, randomized controlled trial was performed at the Faculty of Medicine Siriraj Hospital, Mahidol University, Bangkok, Thailand. We recruited participants, conducted the intervention, and collected the data between June 2021 and October 2021. The study protocol was approved by Siriraj International Review Board (SIRB) (Certificate of approval no. 226/2021). This study was pre-registered in the Thai Clinical Trial Registry (thaiclinicaltrials.org, registration number TCTR20210416001). All participants gave online consent before participating in the study.

### 2.1. Participants

PL posted a study advertisement on a Facebook page to Facebook users across Thailand who had careers related to medicine by using the “Boost Post” feature of Facebook. Anyone interested in participating could scan the QR code on the post or click a link to access the online application form. Then, PL checked the eligibility of those who applied. Eligible participants were physicians, dentists, pharmacists, nurses, practical nurses, medical technologists, physical therapists, traditional medical practitioners, or public health professionals aged 18 years or older that could use the LINE, the popular mobile phone application in Thailand used for online chatting and sending pictures or videos among users. Participants were required to be able to communicate in Thai, have a Stress Test Questionnaire (ST-5) score of ≥5, and a Thai version of the Copenhagen Burnout Inventory (T-CBI) score of ≥50. Applicants were excluded if they had practiced mindfulness at least five days per week for more than one year, were receiving psychotherapy, started treatment for psychiatric illness during the last three months, started taking a new psychotropic medication during the previous three months, or had an existing psychotropic medication dose adjusted during the previous three months.

Participants were withdrawn from the study after enrollment if they (1) participated in another mindfulness program, (2) began psychotherapy, (3) started treatment for a new psychiatric disease, (4) started taking a new psychotropic medication, or (5) had an existing psychotropic medication dose adjusted. Before beginning the study, all participants were informed that they were able to withdraw at any time.

Next, PL emailed eligible applicants the online consent form. The applicants who gave consent to participate then completed the baseline characteristics online questionnaire via email. PL stratified participants into four groups according to their ST-5 and T-CBI scores as follows: low stress and low burnout (ST-5 score < 9, T-CBI score < 61), low stress and high burnout (ST-5 score < 9, T-CBI score ≥ 61), high stress and low burnout (ST-5 score ≥ 9, T-CBI score < 61), and high stress and high burnout (ST-5 score ≥ 9, T-CBI score ≥ 61). After that, PL created an anonymous list of participant ID numbers for each stratum. The list did not contain any other data of participants. NV generated random sequences using permuted block randomization with a block size of four. The possible arrangement within blocks were ABAB, BAAB, BBAA, AABB, ABBA, and BABA. Four series of blocks were created using an online randomization tool. Then, NV applied the random sequences to the list of participant ID numbers in each stratum. Participants were allocated to either Group A or Group B equally according to the randomization. Then, PL emailed participants to inform them of their assigned group and sent links for accessing the online platform used for delivery of interventions.

### 2.2. Online Platform

LINE, a popular mobile phone application in Thailand using for chatting and sending pictures or videos, was used to deliver interventions. We created two chat rooms named ‘Group A’ and ‘Group B’. After participants clicked the emailed link, they automatically joined the chat room corresponding to their assigned group. We used this online platform to send or receive messages from participants and to send online questionnaires and audio files. Participants were not able to know who the other participants were in their group or the other group in order to protect their confidentiality. They were not able to chat with other participants. Each participant could communicate with only the therapist. Conversely, the therapist could send messages, links, or audio files to participants as a group or individually. The therapist’s conversations with participants were unknown to other participants.

### 2.3. Therapist

The single therapist (NV) in this study is a psychiatrist, lecturer and facilitator with six years of experience in mindfulness practice who completed a mindfulness course in Thailand called the Human Work Course. The course teaches a meditation technique known as Dynamic Meditation developed by Luangpor Teean Jittasubho, a well-known Thai Buddhist monk. Dynamic Meditation has practitioners create hand movements or walk, then pay attention to ever-changing body movement, thoughts, or emotions arising in the present moment with kindness and a nonjudgmental attitude. The therapist was not formally trained in any mindfulness-based approaches (they are not available in Thailand at the time).

### 2.4. Mindful Senses (MS) Program

Mindful Senses is a minimal therapist-guided, online audio-based four-week mindfulness program developed by NV. The program is 28 days in duration, including daily messages regarding practical points in mindfulness practice, guided mindfulness practice audio files for self-practice, feedback of practice statistics, and included therapist responses to individual participants’ inquiries. The messages were sent to participants at 9:00 a.m. every day from day 1–28 of the program. Besides practical points in mindfulness practice, the daily messages addressed how mindfulness is associated with burnout and stress. For example, mental rest can be evoked anytime by paying attention to the present moment experience non-judgmentally, which helps reduce burnout and stress. Being aware of thoughts without being caught up in them, accepting them, and letting them come and go can prevent burnout and stress from being excessively amplified by rumination. The contents of the daily messages are provided in Daily Messages S1.

There were four guided mindfulness practice audio files created and recorded by NV. The first, second, and third audio files guided participants to practice mindfulness by using body sensations, surrounding sounds, and front images (what a listener sees) as objects of attention, respectively. The fourth audio file guided participants to use all previous attentional objects as the object of attention. They were 8:22, 8:37, 11:42, and 13:34 min in length, respectively. The four audio files guided participants to let go of any thoughts and emotions arising at the moment and return their attention to the objects of attention. The instructors in the audio files periodically remind participants of their attentional objects to promote awareness in wandering mind, thoughts, and emotions. The English translation of each audio file is provided in Audio Scripts S1.

Participants were required to listen to the mindfulness audio files at least three times per day whenever they were convenient and to follow the guide in the audio files throughout the four weeks of the MS program. The first, second, third, and fourth audio files were sent to participants via the LINE application on days 1, 6, 11, and 16 of the program, respectively. Participants had to listen to the first audio file on days 1–5, the second audio file on days 6–10, the third audio file on days 11–15, and the fourth audio file on days 16–20 of the program at least three times a day. Participants were not able to access the audio files next to what they were assigned. However, they were able to listen to previously completed audio files if they had already listened three times to the audio file assigned for that day. From days 21–28, they were allowed to listen to any of the audio files as they preferred at least three times a day in total. From day 1 to day 28, we recorded how frequently and for how long participants listened to the daily audio files. We sent their audio listening statistics to participants every four days. From days 29 to day 56, we continued to collect listening statistics but did not give feedback on listening statistics to participants.

Participants were able to send inquiries regarding mindfulness practice to the therapist via the LINE application at any time during the four-week program and received a reply within one day. After the 28th day of the program, participants were still able to access all audio files and the 28 daily messages for another four weeks. However, they were not able to ask any mindfulness-related questions of the therapist, and no further daily messages were sent after day 28 of the program.

### 2.5. Psychological Self-Help Article (PSA)

NV wrote the four PSAs used in this study. The topics of PSA included burnout syndrome, stress management, relationship management, and mental health promotion. The English version of the PSA contents (translated from Thai) are provided in PSA Contents S1. There were no mindfulness-related contents in the PSAs. The online PSAs were sent to both groups via the LINE application on day 1 (PSA1), day 8 (PSA2), day 15 (PSA3), and day 22 (PSA4) of the study.

### 2.6. Procedure

Both groups received four weekly PSAs via the LINE application since the beginning of the study. However, each group attended the MS program at different times. Group A participated in the MS program during weeks 1–4 of the study, while Group B participated in the MS program during weeks 9–12. All participants in Group B were asked again before the beginning of the MS program whether they still would like to attend the MS program. Anyone who preferred not to continue the program was withdrawn from the study. The participants and therapist were not blinded to the interventions.

### 2.7. Outcomes

#### 2.7.1. Primary Outcomes

The Thai version of the Copenhagen Burnout Inventory (T-CBI) [27] is a self-reported 19-item questionnaire with five response categories. Possible scores for each item were 0, 25, 50, 75, and 100 (0 = never/almost never or to a very low degree, 100 = always or to a very high degree, reversed score for one item). It has three subscales: personal burnout (six items), work-related burnout (seven items), and client-related burnout (six items). Scores of 50–74, 75–99, and 100 are considered moderate, high, and severe burnout, respectively [28]. The Cronbach’s alpha coefficient for the total scale was 0.96, and 0.91, 0.93, and 0.88 for the three subscales, respectively. T-CBI has high stability within two weeks with an overall Interclass Correlation Coefficient (ICC) value of 0.86, and an ICC value of 0.82 for personal burnout subscale, 0.83 for work-related burnout subscale, and 0.80 for client-related burnout subscale. T-CBI score has a moderate correlation with the Thai version of Maslach Burnout Inventory score (r = 0.51).

According to the developer of the CBI [29], the term “client” is a broad concept that can be adapted to an appropriate term for the respondents when the CBI is used in practice. We replaced the term “client” with “colleague”, considering that medical personnel are the focus of this study, a widely varying numbers of patient exposures during the COVID-19 pandemic, and research evidence [30] showing “perceived support by friends” as a predictor of all components of burnout during the COVID-19 pandemic.

The Stress Test Questionnaire (ST-5) [31] is a five-item self-reported questionnaire. Each question score ranges from 0 to 3 (0 = almost never, 3 = always). It has good internal consistency (Cronbach’s alpha: 0.85). The ST-5 score has moderate correlation with the Hospital Anxiety and Depression Scale (HADS)-anxiety subscale (r = 0.58) and HADS-depression subscale (r = 0.59).

#### 2.7.2. Secondary Outcomes

The Thai version of HADS (Thai HADS) [32] is a 14-item self-reported questionnaire with seven items measuring anxiety and the other seven items measuring depression. Each item has four choices, and the score ranges from 0 to 3. At the cut-off point of ≥ 11, HADS-anxiety subscale has 100.0% sensitivity, 86.0% specificity, a positive predicted value of 0.59 and a negative predicted value of 1.0. In contrast, the HADS-depression subscale has 85.7% sensitivity, 91.3% specificity, a positive predicted value of 0.75, and a negative predicted value of 0.96. Both subscales have good internal consistencies with Cronbach’s alpha coefficient of 0.86 for the anxiety subscale and 0.83 for the depression subscale.

The Thai version of the Philadelphia Mindfulness Scale (PHLMS_TH) [33] is a 20-item self-reported questionnaire with 10 items measuring awareness and the other 10 measuring acceptance. Each item is scored from 1 to 5 (1 = never, 5 = very often). It has good internal consistency for the awareness subscale (Cronbach’s alpha = 0.87) and the acceptance subscale (Cronbach’s alpha = 0.88). It has excellent one-week test-retest reliability for both the awareness subscale (Pearson’s correlation = 0.88) and the acceptance subscale (Pearson’s correlation = 0.89).

The Thai abbreviated version of World Health Organization quality of life (WHOQOL-BREF-THAI) [34] is a 26-item self-reported questionnaire including 24 items for four domains (physical, psychological, social, and environmental), one item for general quality of life, and one item for health-related quality of life. Each item is scored from 1 to 5 (1 = none, never, very poor, very dissatisfied 5 = extremely, complete, always, very good, very satisfied). It has excellent internal consistency for the whole scale (Cronbach’s alpha = 0.90). Cronbach’s alpha of physical domain was 0.73, psychological domain was 0.81, social domain was 0.61, and environmental domain was 0.72.

We assessed feasibility by developing a program feedback questionnaire that included six items. Four items were Likert Scale questions and scored from 1 to 5, asking: (1) How useful do you think the Mindful Senses program is for you (including guided mindfulness practice audio files, therapist response, and daily messages regarding practical points in mindfulness practice)? (1 = not at all useful, 5 = extremely useful); (2) How useful do you think the four psychological self-help articles are for you? (1 = not at all useful, 5 = extremely useful); (3) How difficult or easy do you think it is to use the intervention platform? (1 = very difficult, 5 = very easy); and, (4) How satisfied are you with the overall program of this study (including guided mindfulness practice audio files, psychological self-help articles, intervention platform, therapist, duration of the program, and design of the program)? (1 = very dissatisfied, 5 = very satisfied). The other two items were open-ended questions asking: (1) Do you have any suggestions for improving the program? and, (2) What were the obstacles that kept you from listening to the mindfulness audio files at least three times a day?

### 2.8. Time of Assessment and Incentive

We sent participants the links to online questionnaires via the LINE application. Both groups completed baseline surveys assessing participant baseline characteristics, T-CBI, ST-5, Thai HADS, PHLMS_TH, and WHOQOL-BREF-THAI before the study commencement (T0). After the beginning of the study, Group A had to complete the other two sets of questionnaires at the end of week 4 (T1) and 8 (T2), assessing T-CBI, ST-5, Thai HADS, PHLMS_TH, and WHOQOL-BREF-THAI scores. While, Group B had to complete the other four sets of questionnaires at the end of week 4 (T1), 8 (T2), 12 (T3), and 16 (T4), assessing T-CBI, ST-5, Thai HADS, PHLMS_TH, and WHOQOL-BREF-THAI scores.

Group A completed the questionnaire asking for feedback on MS program usefulness, PSA usefulness, user-friendliness of the platform, and overall satisfaction at the end of week 4 (T1). Group B completed the questionnaire asking for feedback on PSA usefulness at the end of week 4 (T1), and on MS program usefulness, user-friendliness of the platform, and overall satisfaction at the end of week 12 (T3). Each participant received 300 baht (approximately 9 USD) for each set of questionnaires completed (three sets for Group A and five sets for Group B in total).

### 2.9. Adherence

To measure adherence to the research protocol, we programmed the LINE application to automatically record how frequently and for how long participants listened to audio files each day. Group A audio listening statistics were recorded during weeks 1–8, and Group B audio listening statistics were recorded during weeks 9–16.

### 2.10. Withdrawal Criteria Assessment

To assess withdrawal criteria, we sent online questionnaires to participants via the LINE application at T1, T2 (for both groups), T3, and T4 (for Group B), assessing whether participants participated in another mindfulness program, started receiving psychotherapy, started receiving treatment for a new psychiatric disease, started taking a new psychotropic medication, or had an existing psychotropic medication dose adjusted during the previous four weeks.

### 2.11. Statistical Analyses

The estimated sample size needed to compare two means was calculated [35]. We inserted means and SDs from a study by Ireland, MJ et al. [36] into the equation and set α and β values as 0.05 and 0.2, respectively. The calculated optimal sample size was 74. We added 21.6% more sample size to compensate for dropout and arrived at a required sample size of 90.

Baseline characteristics were compared between groups by Pearson’s chi-square test, Fisher’s exact test, or likelihood ratio for categorical variables, independent *t*-test for normally distributed continuous variables, and the Mann-Whitney U test for non-normally distributed continuous variables.

The repeated-measures ANOVA was used to analyze the differences in T-CBI, ST-5, Thai HADS, PHLMS_TH, and WHOQOL-BREF-THAI mean scores across time within group for each group. The within-subject analyses were performed under three (T_0_–T_2_) and five (T_0_–T_4_) time points for group A and B, respectively. Group x Time interaction of T-CBI, ST-5, Thai HADS, PHLMS_TH, and WHOQOL-BREF-THAI mean scores were explored between groups A and B during T_0_–T_2_ to compare the effect of “MS + PSA” with “only PSA”. The two-tailed independent samples *t*-test was used to analyze the differences in T-CBI, ST-5, Thai HADS, PHLMS_TH, and WHOQOL-BREF-THAI mean scores between groups. The mean outcome scores of Group A at T_0_ were compared with that of Group B at T_0_, Group A at T_1_ with Group B at T_3_, and Group A at T_2_ with Group B at T_4_, in order to compare the effect of “MS + PSA” with “MS after PSA”.

We performed the intention-to-treat (ITT) and per-protocol (PP) analyses for within-group and between-group comparisons of T-CBI, ST-5, Thai HADS, PHLMS_TH, and WHOQOL-BREF-THAI mean scores. There was no missing data or data imputation in this study. The *p*-value of <0.05 was used to define statistical significance. To avoid inflated type I errors due to multiple comparisons, the Bonferroni correction was used to adjust the *p*-value for the primary outcomes (CBI-total and ST-5 scores). Original *p*-values of primary outcomes were multiplied by six (two outcomes x three time points) when primary outcomes were compared between groups. Cohen’s d statistic was used for the effect size calculation. Cohen’s d values of ≥ 0.2, 0.5, and 0.8 were interpreted as small, medium, and large effect size, respectively [37].

Descriptive statistics were used to describe guided mindfulness practice audio file listening statistics of participants and their feedback on the program, including usefulness of MS and PSA, difficulty or ease in using the intervention platform, and satisfaction with the program. Pearson’s correlation (for normally distributed variables) and Spearman’s correlation (for non-normally distributed variables) were used to analyze the correlation between audio listening statistics (including total minutes, times of audio listening sessions, and number of days ≥ 3 listening to audio sessions) and the outcomes (including T-CBI, ST-5, Thai HADS, PHLMS_TH, and WHOQOL-BREF-THAI scores).

All statistical analyses were performed using IBM SPSS Statistics for Windows, version 26.0 (IBM Corp., Armonk, NY, USA).

## 3. Results

Two hundred and fifty-nine online applicants were assessed for eligibility (Figure 1), 109 met the inclusion criteria, and 19 were excluded due to regular mindfulness practice (n = 2), starting treatment for psychiatric illness (n = 3), having a psychotropic medication dose adjusted during the last three months (n = 2), or decline to participate (n = 12). The remaining 90 consenting participants were then randomly assigned to Group A (MS + PSA; n = 45) and Group B (MS after PSA; n = 45). There were no statistically significant differences in baseline characteristics between the groups (Table 1), including age, sex, marital status, number of children, education, occupation, type of workplace, income sufficiency, race, religion, domicile, people that the participants are living with, number of total and psychiatric comorbidities, number of current medications, frequency of exercise, sleep hours, history and frequency of substance use, and baseline level of burnout, stress, anxiety, depression, mindfulness, and quality of life.

### 3.1. Primary Outcomes

The burnout and stress level were compared within-group, between MS + PSA and only PSA, and between MS + PSA and MS after PSA by ITT and PP analysis.

#### 3.1.1. Within-Group Comparison

Both Group A (MS + PSA) and Group B (MS after PSA) demonstrated statistically significant within-group improvement in overall burnout, personal-related burnout, colleague-related burnout, and stress (Table 2). The Group A mean work-related burnout score had no statistically significant change (*p* = 0.942) after MS + PSA. The Group B mean work-related burnout score statistically significantly increased after PSA (adjusted *p*-value < 0.001) and decreased after MS (adjusted *p*-value = 0.024). There were no statistically significant differences in work-related burnout scores between baseline, week 12, and week 16 (adjust *p*-value = 1.000 for all three pairwise comparisons). The findings from the PP analysis corresponded to those from the ITT analysis.

#### 3.1.2. MS + PSA Compared with Only PSA

MS + PSA (Group A) showed statistically significant better performance than only PSA (Group B) in reduction of overall burnout, personal-related burnout, work-related burnout, colleague-related burnout, and stress with medium-to-very large effect sizes (d = 1.33, 1.42, 1.13, 0.66, 1.42, respectively) immediately postintervention (T1) (Table 3). The differences in the outcomes remained at one-month follow-up (T2) with medium-to-large effect sizes (d = 0.84, 0.93, 0.82, 0.49, 0.98, respectively). The differences in outcomes between groups at T1 and T2 by PP analysis were consistent with those from the ITT analysis, both for the direction and size of outcome differences.

#### 3.1.3. MS + PSA Compared with MS after PSA

MS + PSA had a statistically significant reduction in personal-related burnout and stress more than MS after PSA immediately post-MS (d = 0.58 and 0.57, respectively) (Table 4). However, the superiority of MS + PSA failed to reach statistical significance at one-month post-MS follow-up. MS + PSA and MS after PSA had no statistically significant difference in their capacity to reduce overall burnout, work-related burnout, and colleague-related burnout. PP analysis revealed a similar pattern as the ITT analysis on the outcome differences between the two intervention sequences, but found statistical significance for personal-related burnout only.

### 3.2. Secondary Outcomes

The anxiety, depression, mindfulness, and quality of life (QOL) levels were compared within-group, between MS + PSA and only PSA, and between MS + PSA and MS after PSA by ITT and PP analysis.

#### 3.2.1. Within-Group Comparison

Both Group A (MS + PSA) and Group B (MS after PSA) demonstrated statistically significant within-group improvement over time in anxiety, depression, mindfulness, awareness, acceptance, overall QOL, and physical, psychological, social, and environmental domain of QOL (Table 2). The findings from the PP analysis corresponded to those from the ITT analysis.

#### 3.2.2. MS + PSA Compared with Only PSA

MS + PSA (Group A) showed statistically significant better performance than only PSA (Group B) in improvement of anxiety, depression, mindfulness, awareness, overall QOL, and the physical, psychological, social, and environmental domain of QOL with medium-to-large effect sizes (d = 1.04, 1.14, 0.70, 0.90, 1.03, 0.91, 0.95, 0.89, and 0.58, respectively) immediately postintervention (T1) (Table 3). The differences in the outcomes remained at one-month follow-up (T2) with medium-to-large effect sizes (d = 0.73, 0.73, 0.66, 0.56, 0.94, 0.84, 0.92, 0.75, and 0.54, respectively). No statistically significant differences between groups were found in the increment of acceptance level. The findings regarding difference in the outcomes between groups at T1 and T2 by the PP analysis corresponded to those from the ITT analysis, both for the direction and size of outcome differences.

#### 3.2.3. MS + PSA Compared with MS after PSA

MS + PSA had statistically significant improvement in anxiety, depression, mindfulness, and the social domain of QOL more than MS after PSA at immediate post-MS (d = 1.13, 0.44, 0.50 and 0.43, respectively). However, MS + PSA remain superior to MS after PSA only in anxiety reduction (d = 0.84) at one-month post-MS follow-up. MS + PSA and MS after PSA had no statistically significant difference in capacity to improve awareness, acceptance, overall QOL, and the physical, psychological, and environmental domain of QOL. Except for depression, the PP analysis revealed a similar pattern to the ITT analysis regarding the outcome differences between the two types of intervention arrangement.

#### 3.2.4. Program Feedback

A majority of participants (47/90; 52.2%) rated the usefulness of the MS program as 4.0 out of 5 (mean 4.23 ± 0.65), the usefulness of PSA as 4.0 out of 5 (n = 49 (54.4%), mean 3.86 ± 0.66), user-friendliness of the platform as 4.0 out of 5 (n = 42 (46.7%), mean 4.17 ± 0.74), and overall program satisfaction as 5.0 out of 5 (n = 42 (46.7%), mean 4.34 ± 0.69) (Table 5). ## participants commented that the program was well-designed, the audios and articles were easy to understand, the therapist was skillful, and that they could gradually practice mindfulness without feeling uncomfortable. They wished this program to be available in public for those who were under stress or interested in mindfulness practice.

Suggestions for program improvement included: (1) the length of the audio files should not exceed 10 min (however, some participants preferred longer audio files); (2) the audios should be playable on computers or in other applications, not just in the LINE application; (3) the audios should be able to be listened to even when the mobile phone screen was turned off; (4) multiple notifications per day could better help remind of upcoming audio listening sessions; (5) might move slower from one audio file to another; (6) might develop more varied audio files to make listening less boring; (7) might give feedback on the listening statistic every day in order to know whether the audio had already been listened to three times; (8) might put together every day messages regarding the practical points in mindfulness practice into one file and send to participants at the end of the MS program for them to keep.

The reported obstacles to complying with the need to listen three times per day to audio included: (1) work overload due to the COVID-19 pandemic made it difficult to have enough available time to listen to the audios three times a day; (2) forgot to listen; (3) urgent work or phone interrupted while listening to the audio; (4) technical problems of mobile phone; (5) exhausted from work; (6) difficulty in finding a place where no one would interrupt while listening; (7) laziness; (8) family responsibilities; (9) had no earphones at the moment; (10) audio files were too long; (11) slow speed or unstable internet; (12) practice notification message were not sent during the available time; and (13) infrequent use of mobile phone.

### 3.3. Adherence

The audio listening statistics of both groups are presented in Table 6. Group A showed higher scores in duration and frequency of audio listening sessions than Group B both when required (while attending the MS program) and not (after finishing the MS program). Ten (22.2%) of 45 participants in Group A listened to the audio files ≥ 3 times per day for more than 60% of the total 28 days of the MS program. Five (11.1%) of 45 participants in Group B were able to follow the requirement. It should be noted that some participants from both groups listened to the audio files more than three times per day, which made the total number of audio listening sessions during the MS program of some participants exceed the 84 times required by the program (three times a day for 28 days).

### 3.4. Correlation between Adherence and Outcomes

Although different audio listening statistics between groups possibly affected the outcome differences between groups, we found no correlations between protocol adherence during the MS program and all outcomes for both Group A and B (Appendix A). However, all participants’ (Group A + B) adherence was found to be negatively correlated with personal-related burnout and positively correlated with the physical domain of QOL (Appendix A). Total eight-week adherence (during MS program + one-month follow-up) of Group A was negatively correlated with mindfulness and acceptance level (Appendix A). In contrast, total eight-week adherence of Group B was positively correlated with mindfulness and acceptance level (Appendix A). It was also positively correlated with overall QOL and the physical and social domains of QOL. Nevertheless, only the physical and social domains of QOL were correlated with all participants’ adherence over the eight weeks (Appendix A).

### 3.5. Therapist Responses to Participant Inquiries

For Group A, seven (15.6%) participants asked questions regarding mindfulness practice at least once during the MS program. Among the seven participants, the total number of questions per participant ranged from one to eight (median (IQR) = 2 (1, 6)). For Group B, only four (8.9%) participants asked questions regarding mindfulness practice at least once during the MS program. Among the four participants, the total number of questions per participant ranged from one to five (mean ± SD = 2.75 ± 1.71).

### 3.6. Attrition

All participants completed all questionnaires and no participants were lost to follow-up. Four (4.4%) participants dropped out of this study. One participant of Group A (MS + PSA) was excluded from the PP analysis because the dosage of the participant’s psychotropic medication was adjusted (Figure 1). Three participants of Group B (MS after PSA) were excluded from the PP analysis because of dosage adjustment of psychotropic medications (n = 2) and attending other mindfulness program (n = 1).

### 3.7. Adverse Effects

We did not systematically measure adverse effects of the program. No participant reported any adverse effects during the study.

## 4. Discussion

We explored the efficacy of the MS program for burnout and stress reduction in medical personnel and found that it decreased burnout and stress levels of medical personnel with statistical significance and large effect sizes, and the effects remained at the one-month follow-up. The size of primary outcomes of this study were larger than other published studies of MBIs. A previous meta-analysis [38] of 38 RCTs showed that MBIs had a statistically significant but small effect on burnout reduction and a statistically significant moderate effect on stress reduction in healthcare professionals and trainees at post-intervention, compared with passive and active controls. At follow-up, a statistically significant effect of MBIs was found only on stress reduction, but not on burnout reduction, and with a small effect size. To date, which factors moderate the effects of MBIs remains unclear [39]. Evidence from a systematic review of meta-analysis of 44 RCTs showed that the associations between the size of outcomes and type of measures (e.g., self-reports), quality of studies, intervention length, age, gender, country of studies, or comparison type were equivocal. However, a meta-analysis [40] found a small but statistically significant association between self-reported home practice and positive intervention outcomes. Another meta-analysis [18] reported that the effect size of outcomes were higher for guided online MBIs than for unguided online MBIs. Therefore, we hypothesize that our larger size of primary outcomes is probably associated with the MS program format that primarily emphasizes daily home practice and incorporation of mindfulness practice into daily life. In addition, the availability of a therapist who could respond within 24 h to participant inquiries may be another important factor contributing to the larger outcomes.

We observed that all three domains of burnout (personal-, work-, and colleague-related burnout) improved after the MS program and the effects remained after one month. This was consistent with a previous study [41] that reported hybrid telephonic MBSR (six-weekly group telephonic MBSR and two full-day in-person retreat) improved all domains of burnout, and the results were maintained at four months of follow-up. Nevertheless, we also found that MS improved personal- and work-related burnout slightly better than colleague-associated burnout. We think that the difference in the size of the outcomes among each domain of burnout might be influenced by other factors contributing to increased burnout level. A meta-analysis [42] reported organization-directed interventions reduced burnout level more than physician-directed interventions. Therefore, healthcare professional burnout might be caused by both individual and organizational factors, and the individual MS intervention alone probably could not reduce all aspects of burnout.

It is worth mentioning that the mean work-related burnout score of Group A had no statistically significant change over time, while that of Group B had a statistically significant increase at weeks four and eight. We think that the increased work-related burnout score in Group B is possibly due to the increasing workload of medical personnel during the COVID-19 pandemic [43]. Yet, the mostly stable work-related burnout score of Group A, despite facing work overload, suggests that the MS program might strengthen participants’ resilience toward challenges in life, as reported in previous studies that MBIs can increase resilience [38].

The MS program provided statistically significant depression and anxiety reductions for medical personnel. The post-intervention effect sizes were large and were slightly reduced one month later. The size of the outcomes in this study was larger than previous studies of its kind. Ghawadra et al. conducted an RCT [44] with a large sample size studied effects of an online four-week guided self-practice mindfulness-based training after a two-hour mindfulness-based training workshop on ward nurses’ anxiety and depression. The results showed a small effect size for anxiety reduction, but no statistically significant effect for depression reduction. We propose that the MS program yielded a larger depression and anxiety reduction for many reasons. First, MS sent a message regarding essential points in mindfulness practice to participants every day which also functioned as a practice reminder, compared with twice-weekly reminders in the study by Ghawadra et al., Second, the Ghawadra et al., study did not assign participants to practice three times a day. The compared study only encouraged participants to practice at their own convenience, giving an unequal sense of responsibility for participants to practice compared with our study. Third, the Ghawadra et al., study did not provide a therapist to respond to participants’ inquiries. A meta-analysis [38] reported that an electronic-delivery facilitator was the most effective type of facilitator compared to a face-to-face trained or student instructor.

The MS program increased overall trait mindfulness of medical personnel with statistical significance. Nevertheless, although MS improved participants’ mean scores in the PHLMS-awareness subscale, it did not improve the PHLMS-acceptance subscale. These counterintuitive results may be explained by the recent debate regarding the validity of self-reports of mindfulness. The validity of self-reported mindfulness has been criticized for its dependency on the respondents’ level of experience in mindfulness practice and their item interpretations. A 2022 study by Hadash Y et al. [45] reported that participants with previous mindfulness practice had statistically significantly lower mindfulness scores than participants without previous practice on some subscales. Moreover, participants with higher performance on behavioral measures of mindfulness provided relatively more accurate and valid self-reports on their mindfulness levels than participants with low mindfulness skills. A recent study [46] explored the validity of the PHLMS and found that the awareness score might be better than the acceptance score in distinguishing between experienced meditators and nonmeditators. Even though the mean acceptance score change in our study did not statistically significantly differ between groups, the scores had statistically significant increases from baseline for both groups. We hypothesize that the PSA that Group B read might promote acceptance level to some degree, leading to non-different acceptance scores between groups. A 2022 meta-analysis [47] supports our hypothesis reporting that self-reported mindfulness is non-unique to mindfulness intervention and can be increased by non-mindfulness interventions, even in a wait-list control group that received no intervention. Another considerable possible explanation of non-different acceptance level between groups is that an acceptance attitude might appear more in those who practice mindfulness for prolonged periods of time. A 2020 study by Morgan, MC et al. [46] reported that years of mindfulness practice correlated with both PHLMS acceptance and awareness, while number of minutes meditating per session or weekly frequency of meditation were not correlated.

The participants’ QOL was largely improved in all domains, including physical, psychological, social, and environmental domains. The effect sizes were large for all domains, except the environmental domain, which showed a medium effect. A previous RCT [48] studied the effect of an eight-week weekly face-to-face meditation program with a practice reminder text message twice a week on nurses’ QOL and found a similar direction of outcomes. The RCT reported improvement in overall QOL and QOL in the physical, psychological, and social domains, but not in the environmental domain. However, another RCT [49] that studied the effect on healthcare providers’ QOL of a four-weekly program with face-to-face one-hour mindfulness sessions found contrasting results. No statistically significant improvement in overall or any individual domain of QOL were found after the four-week mindfulness program compared to the waitlist control. We presume that MS had a statistically significant and larger effect on QOL improvement because the program aimed to incorporate mindfulness practice into a participant’s daily life by having them perform short practice multiple times a day and apply learned techniques to their routine activities. Our presumption was supported by an RCT [50] that found that a higher frequency of mindfulness practice was associated with higher health-related QOL.

Another objective of this study was to explore the outcomes of concurrent MS and PSA, compared with MS after PSA. MS + PSA had better improvement than MS after PSA in personal-related burnout, stress, anxiety, depression, overall mindfulness, and the social domain of QOL. However, at one-month post-MS, MS + PSA showed an advantage above MS after PSA only for anxiety reduction. Although trends of all other outcomes were favoring the concomitant interventions at one-month post-MS, the differences between groups failed to yield statistical significance. The reasons why simultaneous delivery of MS and PSA boosted the effect of anxiety reduction merit exploration. It is worth mentioning that both groups’ outcomes, either immediately or one-month post-MS, were not compared at identical time points. Therefore, the time differences of measurements might affect the study results to some degree. For example, the COVID-19 pandemic situation, which probably affected the measured outcomes [30,43], may have differed in severity at different times of measurement.

Overall, Group A listened to the mindfulness practice guided audio more than Group B, both in duration and number of listening times. Participants in both groups listened to the audios during MS more than post-MS. We think that the differences in duration and number of audio listening sessions between groups, and between MS and post-MS, might be explained by some unexplored factors. A 2021 study by Canby et al. [51] showed that baseline conscientiousness, openness, and depressive symptoms predicted intervention meditation adherence. Hence, we think that our participants’ personality traits may have been different at the baseline and that this affected adherence. However, we could not test our hypothesis because we did not assess participants’ baseline personality traits. Moreover, the depression level of Group B was alleviated to some degree by the PSA before the beginning of MS, which may have led to lower adherence to audio listening sessions in Group B. The Canby et al., study [51] also reported that conscientiousness and depressive symptoms predicted post-intervention adherence. We think that the lower number of audio listening sessions after MS than during MS in both groups might be because their depression had already improved. Finally, some participants might have been able to practice mindfulness in their daily life without a further need for assistance tools such as mindfulness audio files.

All participants’ total audio listening sessions during the MS program inversely correlated only with personal-related burnout and positively correlated only with the physical domain of QOL. This could be interpreted as (1) participants who had lower personal burnout and higher quality of life in the physical domain tended to be willing or able to listen to the audio files, or (2) the more audio files listened to, the more improvement in burnout and quality of life. Nevertheless, the correlations were small and no correlation was found among these variables when each group was analyzed individually. Moreover, no correlation was found between total audio listening sessions during the MS of all participants and the other outcomes. This contrasted with a 2017 meta-analysis [40] that found a small positive correlation between self-reported home mindfulness practice and intervention outcomes. The findings in this study could be interpreted as (1) only a small amount of listening to audio files was enough to produce the same effect as a large amount of listening to audio files; (2) other covert factors that were not controlled (e.g., personality traits and cognitive ability) might also affect the outcomes and confound the correlation between the amount of mindfulness practice and outcomes; or (3) the listening statistics did not reflect the actual amount of participants’ mindfulness practice in real life because they could not measure the amount of self-practice in daily activities without guided audio.

There was a small positive correlation between the physical and social domains of QOL and total listening to audio files of all participants from the beginning of MS to the one-month follow-up. No correlation was found between all participants’ total audio listening sessions during the two-month period and the other outcomes. The findings could be explained as (1) participants who had higher physical or social QOL tended to be willing or able to listen to the mindfulness audio slightly more, or (2) participants who listened to the mindfulness audio more tended to have slightly better improvement in physical and social QOL. Surprisingly, the total amount of audio listening sessions from the beginning of MS to the one-month follow-up had a medium positive correlation with acceptance for Group B. In contrast, it had a medium inverse correlation with acceptance for Group A. We suggest some possible explanations. First, the higher number of mindfulness audio listening sessions could contribute to increased acceptance, or it could be a consequence of low acceptance by participants who had low mindfulness skills and needed to use more of the audio as a mindfulness practice-assistance tool. Second, the PHLMS probably had some limitations in acceptance-level measurement. All items in the acceptance subscale were solely reverse-scored [33,52], which might affect participants’ item interpretation to some degree. It also gives lower acceptance scores to those who do the distraction technique more often. Although trying to distract oneself from negative emotions or experience can be viewed as reluctance to accept the present moment experience, experienced meditators who have high levels of acceptance can still choose to direct their attention to other objects mindfully. They do not always need to keep paying attention to the negative experience. Furthermore, focused attention meditation techniques that have meditators return their attention to attentional objects after realizing that their minds have wandered can be viewed as a distraction in the views of participants who completed PHLMS. Additionally, it is worth noting that participants who had better self-awareness after practicing mindfulness for some period probably became more aware of their judging mind. They possibly rated their acceptance levels lower than baseline, which did not necessarily mean that their acceptance levels actually decreased.

More than four-fifths of the participants were satisfied with MS and PSA, rated MS as useful, and the platform as user-friendly. Furthermore, participants commented that the MS program was well-developed and the contents were easy-to-understand and practical. Therefore, the program seemed to be acceptable for medical personnel. However, the program may be further developed to better suit healthcare professionals with high workloads. For example, through limiting the length of each audio file to 10 min, multiple practice notifications per day and daily feedback of listening statistics to encourage motivation to practice, developing more mindfulness practice audio files to improve engagement and matching each individual’s preference style, enabling participants to listen to the mindfulness audio despite the smart phone screen being turned-off, etc.

Of 90 participants, four (4.4%) were excluded from the PP analysis because the dosages of their psychotropic medications were adjusted or they attended other mindfulness programs during the study. No participants were lost to follow-up in this study and the dropout rate was lower than previous mindfulness studies. A meta-analysis [53] of RCTs reported the mean dropout rate of mindfulness application intervention was 31.6% (range 0–73.3%). Previous RCTs [54,55] found that the self-guided online MBI group had a dropout rate of 57.3–79.4%. The lower dropout rate in this study was probably because we informed all participants every week via our platform where we were regarding the research procedure and what would be the next steps. Also, we offered close support for both groups in case participants had some technical problems regarding platform use. Participants could communicate their issues directly to PL, via the LINE application or email during the study. A previous RCT that supported our hypothesis found that the MBI group in which participants received welcome emails, inquiry responses, and practice reminder emails had a lower dropout rate than an MBI group without these supports. The control group also received some benefit from active control condition (PSA), probably contributing to fewer dropouts [56]. A previous study [57] that supported our hypothesis reported that increased waitlist time was a predictor of a higher dropout rate.

Despite promising results, some limitations of this study should be noted. First, although randomization and allocation concealment were performed, participants and therapist were not blinded for the group assignment because it was not possible for our study. Second, the outcome measures were all self-reports that are at risk of retrospective recall bias. Moreover, the unblinded design combined with self-report measures made the risk of bias from demand characteristics inevitable. Nevertheless, we tried to mitigate the risk of bias from demand characteristics by informing participants at the beginning of every questionnaire to respond to the items in the questionnaires according to their actual experiences, not what was good or what they thought they should be. Although the validity of self-reports of trait mindfulness were subject to the levels of mindfulness practice experience of participants, we did not measure mindfulness by other methods (e.g., behavioral measures of mindfulness) because we aimed to conduct the research as fully online during the COVID-19 pandemic. Last, our results from health professionals may not be generalizable to other groups. Most of the population enrolled in this study were female and young. Therefore, interpretation of the results for men and older persons should be done cautiously. Additionally, only one therapist (NV) provided the intervention, so the generalizability of the intervention effects outside this provider is unknown.

The results of the present study could be applied in several ways. First, online MBI developers might integrate some essential components of MS (e.g., available therapist for within-day response to participant’s inquiries, close technical support, brief mindfulness practice multiple times a day, etc.) into their program to improve the efficacy of online MBI and lower the dropout rate. Second, the MS program might be used as an alternative intervention for burnout and stress reduction in medical personnel. It is available anywhere and anytime online, suitable for busy healthcare professionals. There is also a therapist available for answering participants’ mindfulness-related inquiries every day, providing intensive coaching that other mindfulness applications do not offer. Moreover, it protects the confidentiality of participants and prevents stigma through its thoughtful design. Additionally, it can considerably reduce burnout and stress level in only a month, making it an attractive choice for healthcare workers to reduce their burnout and stress. Third, online psychological self-help articles could be provided to participants in a waiting list for MBIs. This could help mitigate participants’ symptoms and decrease the chance of dropout or lost-to-follow-up. Finally, outcome-related online psychological self-help articles might be added in an online MBI for boosting the effect.

Several aspects should be considered for further investigation. Despite promising results of the MS program towards burnout and stress reduction in medical personnel, the program effect for male and older medical personnel is yet to be confirmed. Although challenging, blinding participants and therapist when possible will improve the rigor of the study methodology. Also, including objective measures (e.g., EEG, fMRI, blood or saliva tests) in a study can support the validity of subjective reports and lower the risk of bias from demand characteristics. Ecological Momentary Assessment (EMA) [58], the repeated collection of self-report momentary experience, might be used instead of some self-report measures to obtain data with higher validity and lower retrospective recall bias. Although the design of the MS program seemed to improve the size of MBI outcomes, definite mediators of the size of outcomes need to be explored. Further investigation of long-term effects, adverse effects, and cost-effectiveness are warranted. Last but not least, concurrently adding outcome-related psychoeducation articles along with online MBIs may help increase efficacy, and further investigation is necessary to confirm the finding.

## 5. Conclusions

The MS program is the online MBI that largely reduced burnout and stress in medical personnel, and its effects remained at the one-month follow-up. Depression, anxiety, mindfulness and QOL were also improved. Concurrent reading of PSA and attending the MS program boosted the anxiety reduction effect. Participants reported the MS program was beneficial. They were satisfied with the program and saw the platform as user-friendly. The dropout rate was low. The MS program has the potential to be a highly useful tool for medical personnel who are burned out and stressed and unable to attend in-person mental health care because of confidentiality or social stigma concerns and limited time. More research into the program’s long-term impacts, negative effects, and cost-effectiveness could support its wider implementation.

## Figures and Tables

**Figure 1 healthcare-10-02532-f001:**
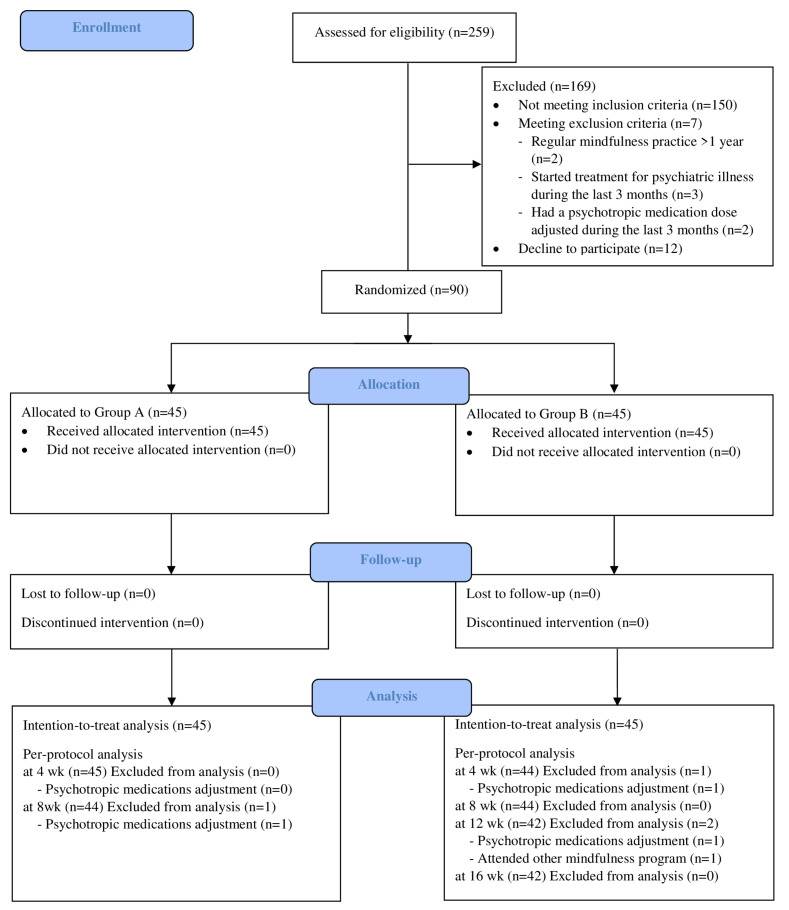
CONSORT flow diagram.

**Table 1 healthcare-10-02532-t001:** Baseline demographic and characteristics of participants.

Baseline Characteristics	Group A n = 45 Mean ± SD Median (IQR) n (%)	Group B n = 45 Mean ± SD Median (IQR) n (%)	*p*
Age, year	34.07 ± 8.49	32.87 ± 7.27	0.473 ^‡^
Range	22–60	25–53	
Female	37 (82.2%)	39 (86.7%)	0.561 ^†^
Marital status			0.615 ^¶^
Married	8 (17.8%)	9 (20.0%)	
Single	34 (75.6%)	35 (77.8%)	
Divorced	2 (4.4%)	1 (2.2%)	
Separated	1 (2.2%)	0 (0.0%)	
Number of children			0.243 ^¶^
0	36 (80.0%)	38 (84.4%)	
1–2	7 (15.6%)	7 (15.6%)	
3–4	2 (4.4%)	0 (0.0%)	
Education			0.791 ^†^
Bachelor	28 (62.2%)	31 (68.9%)	
Master	8 (17.8%)	7 (15.6%)	
Higher graduate diploma	9 (20.0%)	7 (15.6%)	
Occupation			0.156 ^¶^
Medical physician	15 (33.3%)	18 (40.0%)	
Dentist	3 (6.7%)	5 (11.1%)	
Pharmacist	4 (8.9%)	0 (0.0%)	
Nurse	10 (22.2%)	11 (24.4%)	
Medical technician	2 (4.4%)	0 (0.0%)	
Physiotherapist	5 (11.1%)	6 (13.3%)	
Others	6 (13.3%)	5 (11.1%)	
Workplace			0.494 ^†^
Medical school	16 (35.6%)	15 (33.3%)	
Regional hospital	7 (15.6%)	5 (11.1%)	
General hospital	3 (6.7%)	7 (15.6%)	
Community hospital	10 (22.2%)	6 (13.3%)	
Others	9 (20.0%)	12 (26.7%)	
Work hours per week in the last one month	55.82 ± 22.41	57.18 ± 20.80	0.767 ^‡^
Income (baht) per month	40,000 (24,500, 55,000)	40,000 (22,045, 60,000)	0.774 *
Income sufficiency			0.761 ^†^
Enough and savings	27 (60.0%)	24 (53.3%)	
Just enough	13 (28.9%)	14 (31.1%)	
Not enough	5 (11.1%)	7 (15.6%)	
Race			1.000 ^§^
Thai	44 (97.8%)	45 (100.0%)	
Chinese	1 (2.2%)	0 (0.0%)	
Religion			0.474 ^¶^
Buddhism	40 (88.9%)	43 (95.6%)	
Christianity	3 (6.7%)	1 (2.2%)	
Islam	2 (4.4%)	1 (2.2%)	
Domicile			0.288 ^†^
Bangkok	17 (37.8%)	22 (48.9%)	
Other	28 (62.2%)	23 (51.1%)	
Living with			0.124 ^¶^
Alone	9 (20.0%)	7 (15.6%)	
With family	33 (73.3%)	27 (60.0%)	
With friend	2 (4.4%)	8 (17.8%)	
Other	1 (2.2%)	3 (6.7%)	
Number of comorbidities (include psychiatric disease)			0.112 ^¶^
0	30 (66.7%)	28 (62.2%)	
1	10 (22.2%)	16 (35.6%)	
2–3	5 (11.1%)	1 (2.2%)	
Number of psychiatric comorbidities			1.000 ^§^
0	44 (97.8%)	44 (97.8%)	
1	1 (2.2%)	1 (2.2%)	
Number of current regularly taken medications			0.575 ^¶^
0	32 (71.1%)	34 (75.6%)	
1–2	10 (22.2%)	10 (22.2%)	
3–4	3 (6.7%)	1 (2.2%)	
Exercise in the last one month (days/week)			0.955 ^†^
0	18 (40.0%)	18 (40.0%)	
1–2	20 (44.4%)	19 (42.2%)	
3–5	7 (15.6%)	8 (17.8%)	
Sleep in the last one month (hours/day)	6.29 ± 1.53	6.02 ± 1.11	0.348 ^‡^
Substance use			1.000 ^¶^
Never or rarely use	43 (95.6%)	43 (95.6%)	
Past regular use	1 (2.2%)	1 (2.2%)	
Current regular use	1 (2.2%)	1 (2.2%)	
Frequency of substance use in the last 12 months (times/month)			0.556^¶^
0	38 (84.4%)	37 (82.2%)	
1–2	3 (6.7%)	6 (13.3%)	
4–6	3 (6.7%)	1 (2.2%)	
20	1 (2.2%)	1 (2.2%)	
Baseline outcome scores			
T-CBI	62.08 ± 9.43	64.82 ± 11.75	0.224 ^‡^
ST-5	8.87 ± 2.48	8.93 ± 2.37	0.897 ^‡^
Thai HAD-anxiety	9.29 ± 3.02	10.16 ± 2.81	0.162 ^‡^
Thai HAD-depression	8.76 ± 3.28	8.27 ± 3.22	0.477 ^‡^
PHLMS_TH	58.00 ± 6.78	57.56 ± 7.98	0.777 ^‡^
WHOQOL-BREF-THAI	81.38 ± 10.98	82.98 ± 11.49	0.501 ^‡^

Abbreviations: T-CBI, the Thai version of the Copenhagen Burnout Inventory; ST-5, the Stress Test Questionnaire; Thai HAD-anxiety, the Thai version of HADS anxiety subscale; Thai HAD-depression, the Thai version of HADS depression subscale; PHLMS_TH, the Thai version of the Philadelphia Mindfulness Scale; WHOQOL-BREF-THAI, the Thai abbreviated version of World Health Organization quality of life. ^†^ Chi-square test, ^‡^ *t* test, ^§^ Fisher’s exact test, ^¶^ Likelihood ratio, * Mann-Whitney U.

**Table 2 healthcare-10-02532-t002:** Within-group outcome changes over time.

**ITT**										
**Outcomes**	**Group A** **mean ± SD, n = 45**			* **p** *	**Group B** **mean ± SD, n = 45**					* **p** *
	**Baseline**	**4 weeks**	**8 weeks**		**Baseline**	**4 weeks**	**8 weeks**	**12 weeks**	**16 weeks**	
CBI-total	62.08 ± 9.43	38.65 ± 13.56	39.85 ± 16.31	<0.001 ^†^	64.82 ± 11.75	56.35 ± 12.98	53.30 ± 15.62	43.13 ± 13.39	43.16 ± 14.83	<0.001 ^‡^
CBI-personal	66.39 ± 12.45	36.76 ± 15.92	38.98 ± 19.18	<0.001 ^†^	67.22 ± 12.39	59.07 ± 15.55	55.37 ± 16.03	46.48 ± 17.36	45.46 ± 15.96	<0.001 ^‡^
CBI-work	42.14 ± 9.50	41.59 ± 14.69	42.38 ± 16.25	0.942 ^†^	42.06 ± 11.59	58.73 ± 15.74	55.87 ± 16.81	45.00 ± 14.27	45.32 ± 14.80	<0.001 ^‡^
CBI-colleague	53.52 ± 20.58	37.13 ± 20.62	37.78 ± 20.61	<0.001 ^†^	58.61 ± 21.51	50.83 ± 21.07	48.24 ± 21.95	37.59 ± 18.09	38.33 ± 21.09	<0.001 ^‡^
ST-5	8.87 ± 2.48	3.96 ± 1.72	4.13 ± 2.52	<0.001 ^†^	8.93 ± 2.37	6.93 ± 2.42	6.84 ± 2.98	5.11 ± 2.26	5.00 ± 2.51	<0.001 ^†^
HAD-anxiety	9.29 ± 3.02	5.78 ± 2.52	5.80 ± 3.42	<0.001 ^‡^	10.16 ± 2.81	8.87 ± 3.33	8.38 ± 3.64	8.58 ± 2.42	8.31 ± 2.49	0.006 ^‡^
HAD-depression	8.76 ± 3.28	3.89 ± 3.14	4.62 ± 3.59	<0.001 ^†^	8.27 ± 3.22	7.76 ± 3.64	7.27 ± 3.62	5.47 ± 3.96	5.00 ± 3.40	<0.001 ^‡^
PHLMS-total	58.00 ± 6.78	69.27 ± 7.95	66.73 ± 7.84	<0.001 ^†^	57.56 ± 7.98	64.20 ± 6.49	61.40 ± 8.41	65.49 ± 6.99	65.27 ± 7.82	<0.001 ^‡^
PHLMS-awareness	32.73 ± 6.27	38.09 ± 4.38	36.69 ± 4.57	<0.001 ^‡^	31.51 ± 6.51	33.56 ± 5.58	33.84 ± 5.56	36.60 ± 4.58	36.49 ± 5.43	<0.001 ^‡^
PHLMS-acceptance	25.27 ± 6.57	31.18 ± 5.92	30.04 ± 7.17	<0.001 ^†^	26.04 ± 6.19	30.64 ± 5.63	27.56 ± 6.69	28.89 ± 5.99	28.78 ± 6.64	0.001 ^†^
QOL-total	81.38 ± 10.98	96.40 ± 8.81	96.58 ± 10.99	<0.001 ^‡^	82.98 ± 11.49	85.47 ± 12.22	85.98 ± 11.47	94.00 ± 13.10	93.20 ± 14.41	<0.001 ^‡^
QOL-physical	22.91 ± 3.25	27.49 ± 2.83	27.31 ± 2.95	<0.001 ^†^	23.67 ± 3.31	24.56 ± 3.58	24.60 ± 3.49	26.31 ± 3.47	26.02 ± 4.00	<0.001 ^‡^
QOL-psychological	17.89 ± 2.80	21.98 ± 2.81	22.04 ± 2.94	<0.001 ^†^	17.82 ± 2.68	18.84 ± 3.72	18.91 ± 3.78	21.33 ± 3.44	21.22 ± 3.91	<0.001 ^‡^
QOL-social	9.87 ± 1.93	11.49 ± 1.58	11.47 ± 1.94	<0.001 ^‡^	9.87 ± 2.21	9.87 ± 2.03	9.96 ± 2.10	10.71 ± 2.02	10.84 ± 2.07	0.001 ^‡^
QOL-environment	24.87 ± 4.95	28.33 ± 3.30	28.56 ± 4.30	<0.001 ^‡^	25.62 ± 4.65	26.11 ± 4.25	26.36 ± 3.85	28.62 ± 4.90	28.24 ± 4.90	<0.001 ^‡^
**PP**										
**Outcomes**	**Group A** **mean ± SD, n = 44**			** *p* **	**Group B** **mean ± SD, n = 42**					** *p* **
	**Baseline**	**4 weeks**	**8 weeks**		**Baseline**	**4 weeks**	**8 weeks**	**12 weeks**	**16 weeks**	
CBI-total	62.35 ± 9.35	38.73 ± 13.70	39.80 ± 16.49	<0.001 ^†^	64.25 ± 11.53	56.17 ± 13.27	53.64 ± 15.93	42.26 ± 12.32	43.42 ± 15.31	<0.001 ^‡^
CBI-personal	66.86 ± 12.19	36.84 ± 16.10	39.11 ± 19.38	<0.001 ^†^	66.17 ± 11.72	58.04 ± 15.46	55.95 ± 15.86	45.04 ± 16.08	45.24 ± 16.45	<0.001 ^‡^
CBI-work	42.53 ± 9.24	41.88 ± 14.72	42.69 ± 16.30	0.938 ^†^	41.58 ± 11.75	58.08 ± 16.09	55.87 ± 16.80	43.88 ± 13.28	45.41 ± 15.14	<0.001 ^‡^
CBI-colleague	53.13 ± 20.65	36.93 ± 20.82	37.12 ± 20.37	<0.001 ^†^	58.93 ± 21.94	52.08 ± 20.27	48.71 ± 22.44	37.60 ± 17.40	39.29 ± 21.15	<0.001 ^‡^
ST-5	8.91 ± 2.49	3.91 ± 1.71	4.16 ± 2.54	<0.001 ^†^	8.90 ± 2.41	6.88 ± 2.49	6.83 ± 2.85	5.02 ± 2.12	5.07 ± 2.58	<0.001 ^†^
HAD-anxiety	9.36 ± 3.01	5.77 ± 2.55	5.77 ± 3.46	<0.001 ^‡^	10.10 ± 2.79	8.81 ± 3.32	8.40 ± 3.61	8.52 ± 2.43	8.40 ± 2.53	0.014 ^‡^
HAD-depression	8.77 ± 3.31	3.84 ± 3.16	4.64 ± 3.64	<0.001 ^†^	8.14 ± 3.27	7.50 ± 3.62	7.36 ± 3.55	5.26 ± 3.99	5.05 ± 3.48	<0.001 ^‡^
PHLMS-total	58.09 ± 6.83	69.32 ± 8.04	66.70 ± 7.93	<0.001 ^†^	57.48 ± 8.06	64.10 ± 6.32	61.64 ± 8.57	65.33 ± 7.19	65.31 ± 8.07	<0.001 ^‡^
PHLMS-awareness	32.84 ± 6.30	38.11 ± 4.43	36.77 ± 4.58	<0.001 ^‡^	31.36 ± 6.56	33.52 ± 5.52	33.98 ± 5.65	36.57 ± 4.60	36.38 ± 5.60	<0.001 ^‡^
PHLMS-acceptance	25.25 ± 6.64	31.20 ± 5.99	29.93 ± 7.22	<0.001 ^†^	26.12 ± 6.09	30.57 ± 5.79	27.67 ± 6.88	28.76 ± 5.96	28.93 ± 6.82	0.005 ^†^
QOL-total	81.27 ± 11.08	96.48 ± 8.90	96.50 ± 11.11	<0.001 ^‡^	83.50 ± 11.66	86.02 ± 12.43	85.86 ± 11.70	94.57 ± 12.96	93.07 ± 14.82	<0.001 ^‡^
QOL-physical	22.89 ± 3.29	27.50 ± 2.86	27.27 ± 2.97	<0.001 ^†^	23.88 ± 3.32	24.69 ± 3.66	24.52 ± 3.51	26.52 ± 3.34	26.10 ± 4.09	<0.001 ^‡^
QOL-psychological	17.91 ± 2.83	21.95 ± 2.84	22.00 ± 2.96	<0.001 ^†^	17.71 ± 2.66	18.95 ± 3.77	18.69 ± 3.70	21.45 ± 3.42	21.14 ± 4.02	<0.001 ^‡^
QOL-social	9.91 ± 1.93	11.50 ± 1.59	11.48 ± 1.96	<0.001 ^‡^	9.83 ± 2.28	9.76 ± 2.06	9.95 ± 2.16	10.64 ± 2.07	10.79 ± 2.12	0.002 ^‡^
QOL-environment	24.70 ± 4.89	28.39 ± 3.32	28.57 ± 4.35	<0.001 ^‡^	26.02 ± 4.51	26.40 ± 4.21	26.55 ± 3.90	28.88 ± 4.79	28.19 ± 5.01	0.001 ^†^

Abbreviations: ITT, intention-to-treat analysis; PP, per-protocol analysis; CBI, Thai version of the Copenhagen Burnout Inventory; ST-5, the Stress Test Questionnaire; HAD-anxiety, the Thai version of HADS anxiety subscale; HAD-depression, the Thai version of HADS depression subscale; PHLMS, the Thai version of Philadelphia Mindfulness Scale; QOL, the Thai abbreviated version of World Health Organization quality of life (WHOQOL-BREF-THAI). ^†^ Sphericity Assumed, ^‡^ Greenhouse-Geisser.

**Table 3 healthcare-10-02532-t003:** Between-group outcome differences at baseline and the end of weeks 4 and 8.

	ITT			PP		
Outcomes	Mean Difference ^a^ (95% CI)	*p*	*d*	Mean Difference ^a^ (95% CI)	*p*	*d*
CBI-total						
Time × Group		<0.001 ^‡^			<0.001 ^‡^	
Baseline	−2.75 (−7.21, 1.71)	1.000 ^§^	0.26	−2.48 (−7.02, 2.05)	1.000 ^§^	0.23
4 weeks	−17.69 (−23.25, −12.13)	<0.001 ^§^	1.33	−17.79 (−23.47, −12.12)	<0.001 ^§^	1.33
8 weeks	−13.45 (−20.14, −6.76)	<0.001 ^§^	0.84	−13.52 (−20.36, −6.67)	0.001 ^§^	0.84
CBI-personal						
Time × Group		<0.001 ^‡^			<0.001 ^‡^	
Baseline	−0.83 (16.04, 4.37)	0.751	0.07	0.00 (−5.19, 5.19)	1.000	0.00
4 weeks	−22.32 (−28.91, −15.72)	<0.001	1.42	−21.97 (−28.69, −15.25)	<0.001	1.39
8 weeks	−16.39 (−23.79, −8.98)	<0.001	0.93	−16.10 (−23.66, −8.53)	<0.001	0.90
CBI-work						
Time × Group		<0.001 ^‡^			<0.001 ^‡^	
Baseline	0.08 (−4.36, 4.52)	0.972	0.01	0.49 (−3.99, 4.96)	0.829	0.05
4 weeks	−17.14 (−23.52, -10.76)	<0.001	1.13	−16.56 (−23.03, −10.09)	<0.001	1.08
8 weeks	−13.49 (−20.42, −6.57)	<0.001	0.82	−12.91 (−19.94, −5.87)	<0.001	0.78
CBI-colleague						
Time × Group		0.176 ^‡^			0.154 ^‡^	
Baseline	−5.09 (−13.91, 3.73)	0.254	0.24	−5.97 (−14.90, 2.97)	0.188	0.28
4 weeks	−13.70 (−22.44, −4.97)	0.002	0.66	−15.06 (−23.67, −6.44)	0.001	0.74
8 weeks	−10.46 (−19.38, −1.54)	0.022	0.49	−11.65 (−20.61, −2.68)	0.012	0.55
ST-5						
Time × Group		<0.001 ^‡^			<0.001 ^‡^	
Baseline	−0.07 (−1.08, 0.95)	1.000 ^§^	0.03	0.00 (−1.04, 1.04)	1.000 ^§^	0.00
4 weeks	−2.98 (−3.86, −2.10)	<0.001 ^§^	1.42	−3.00 (−3.89, −2.11)	<0.001 ^§^	1.42
8 weeks	−2.71 (−3.87, −1.56)	<0.001 ^§^	0.98	−2.59 (−3.76, −1.43)	<0.001 ^§^	0.94
HAD-anxiety						
Time × Group		0.004 ^‡^			0.003 ^‡^	
Baseline	−0.87 (−2.09, 0.36)	0.162	0.30	−0.68 (−1.90, 0.54)	0.270	0.24
4 weeks	−3.09 (−4.33, −1.85)	<0.001	1.04	−3.00 (−4.25, −1.75)	<0.001	1.01
8 weeks	−2.58 (−4.06, −1.10)	0.001	0.73	−2.50 (−4.00, −1.00)	0.001	0.71
HAD-depression						
Time × Group		<0.001 ^†^			<0.001 ^†^	
Baseline	0.49 (−0.87, 1.85)	0.477	0.15	0.55 (−0.85, 1.94)	0.438	0.17
4 weeks	−3.87 (−5.29, −2.44)	<0.001	1.14	−3.84 (−5.29, −2.40)	<0.001	1.13
8 weeks	−2.64 (−4.16, −1.13)	0.001	0.73	−2.57 (−4.11, −1.03)	0.001	0.71
PHLMS-total						
Time × Group		0.028 ^†^			0.037 ^†^	
Baseline	0.44 (−2.66, 3.55)	0.777	0.06	0.34 (−2.81, 3.49)	0.830	0.05
4 weeks	5.07 (2.03, 8.11)	0.001	0.70	4.96 (1.86, 8.05)	0.002	0.68
8 weeks	5.33 (1.93, 8.74)	0.002	0.66	5.09 (1.64, 8.55)	0.004	0.62
PHLMS-awareness						
Time × Group		0.037 ^‡^			0.057 ^‡^	
Baseline	1.22 (−1.45, 3.90)	0.367	0.19	1.32 (−1.41, 4.05)	0.340	0.20
4 weeks	4.53 (2.43, 6.64)	<0.001	0.90	4.46 (2.32, 6.59)	<0.001	0.88
8 weeks	2.84 (0.71, 4.98)	0.009	0.56	2.82 (0.66, 4.98)	0.011	0.55
PHLMS-acceptance						
Time × Group		0.107 ^†^			0.121 ^†^	
Baseline	−0.78 (−3.45, 1.90)	0.565	0.12	−0.98 (−3.69, 1.73)	0.475	0.15
4 weeks	0.53 (−1.89, 2.95)	0.663	0.09	0.50 (−1.97, 2.97)	0.689	0.09
8 weeks	2.49 (−0.42, 5.40)	0.092	0.36	2.27 (−0.68, 5.23)	0.130	0.33
QOL-total						
Time × Group		<0.001 ^‡^			<0.001 ^‡^	
Baseline	−1.60 (−6.31, 3.11)	0.501	0.14	−1.82 (−6.63, 2.99)	0.454	0.16
4 weeks	10.93 (6.47, 15.40)	<0.001	1.03	10.84 (6.29, 15.39)	<0.001	1.01
8 weeks	10.60 (5.89, 15.31)	<0.001	0.94	10.46 (5.64, 15.27)	<0.001	0.92
QOL-physical						
Time × Group		<0.001 ^‡^			<0.001 ^‡^	
Baseline	−0.76 (−2.13, 0.62)	0.278	0.23	−0.86 (−2.26, 0.53)	0.222	0.26
4 weeks	2.93 (1.58, 4.29)	<0.001	0.91	2.86 (1.49, 4.24)	<0.001	0.88
8 weeks	2.71 (1.36, 4.07)	<0.001	0.84	2.64 (1.26, 4.02)	<0.001	0.81
QOL-psychological						
Time × Group		<0.001 ^†^			<0.001 ^†^	
Baseline	−0.07 (−1.08, 1.22)	0.908	0.02	0.14 (−1.03, 1.31)	0.817	0.05
4 weeks	3.13 (1.75, 4.51)	<0.001	0.95	3.11 (1.70, 4.53)	<0.001	0.94
8 weeks	3.13 (1.71, 4.55)	<0.001	0.92	3.09 (1.64, 4.54)	<0.001	0.90
QOL-social						
Time × Group		<0.001 ^‡^			0.001 ^‡^	
Baseline	0.00 (−0.87, 0.87)	1.000	0.00	0.07 (−0.82, −0.95)	0.878	0.03
4 weeks	1.62 (0.86, 2.38)	<0.001	0.89	1.66 (0.88, 2.44)	<0.001	0.91
8 weeks	1.51 (0.67, 2.36)	0.001	0.75	1.55 (0.68, 2.41)	0.001	0.76
QOL-environment						
Time × Group		0.009 ^‡^			0.005 ^‡^	
Baseline	−0.76 (−2.77, 1.26)	0.458	0.16	−1.02 (−3.04, 1.00)	0.317	0.21
4 weeks	2.22 (0.63, 3.82)	0.007	0.58	2.21 (0.58, 3.83)	0.008	0.58
8 weeks	2.20 (0.49, 3.91)	0.012	0.54	2.16 (0.41, 3.91)	0.016	0.52

Abbreviations: ITT, intention-to-treat analysis; PP, per-protocol analysis; CBI, Thai version of the Copenhagen Burnout Inventory; ST-5, the Stress Test Questionnaire; HAD-anxiety, the Thai version of HADS anxiety subscale; HAD-depression, the Thai version of HADS depression subscale; PHLMS, the Thai version of Philadelphia Mindfulness Scale; QOL, the Thai abbreviated version of World Health Organization quality of life (WHOQOL-BREF-THAI). ^a^ Mean difference = mean outcome score of Group A—mean outcome score of Group B. ^†^ Sphericity Assumed. ^‡^ Greenhouse-Geisser. ^§^ Bonferroni correction was used to adjust *p*-value of primary outcomes (CBI-total and ST-5). Adjusted *p*-value = original *p*-value × 6 (2 outcomes × 3 time points).

**Table 4 healthcare-10-02532-t004:** Between-group outcome differences immediately post-MS * and one-month post-MS **.

	ITT				PP			
Outcomes	Group A n = 45 Mean ± SD	Group B n = 45 Mean ± SD	*p*	*d*	Group A n = 45 ^a^, 44 ^b^ Mean ± SD	Group B n = 42 Mean ± SD	*p*	*g*
CBI-total								
Baseline	62.08 ± 9.43	64.82 ± 11.75	1.000 ^§^	0.26	62.08 ± 9.43	64.82 ± 11.75	1.000 ^§^	0.26
Immediate post-MS	38.65 ± 13.56	43.13 ± 13.39	0.713 ^§^	0.33	38.65 ± 13.56	42.26 ± 12.32	1.000 ^§^	0.28
1-month post-MS	39.85 ± 16.31	43.16 ± 14.83	1.000 ^§^	0.21	39.80 ± 16.49	43.42 ± 15.31	1.000 ^§^	0.23
CBI-personal								
Baseline	66.39 ± 12.45	67.22 ± 12.39	0.751	0.07	66.39 ± 12.45	67.22 ± 12.39	0.751	0.07
Immediate post-MS	36.76 ± 15.92	46.48 ± 17.36	0.007	0.58	36.76 ± 15.92	45.04 ± 16.08	0.018	0.52
1-month post-MS	38.98 ± 19.18	45.46 ± 15.96	0.085	0.37	39.11 ± 19.38	45.24 ± 16.45	0.119	0.34
CBI-work								
Baseline	42.14 ± 9.50	42.06 ± 11.59	0.972	0.01	42.14 ± 9.50	42.06 ± 11.59	0.972	0.01
Immediate post-MS	41.59 ± 14.69	45.00 ± 14.27	0.267	0.24	41.59 ± 14.69	43.88 ± 13.28	0.449	0.16
1-month post-MS	42.38 ± 16.25	45.32 ± 14.80	0.373	0.19	42.69 ± 16.30	45.41 ± 15.14	0.427	0.17
CBI-colleague								
Baseline	53.52 ± 20.58	58.61 ± 21.51	0.254	0.24	53.52 ± 20.58	58.61 ± 21.51	0.254	0.24
Immediate post-MS	37.13 ± 20.62	37.59 ± 18.09	0.910	0.02	37.13 ± 20.62	37.60 ± 17.40	0.909	0.02
1-month post-MS	37.78 ± 20.61	38.33 ± 21.09	0.900	0.03	37.12 ± 20.37	39.29 ± 21.15	0.630	0.10
ST-5								
Baseline	8.87 ± 2.48	8.93 ± 2.37	1.000 ^§^	0.02	8.87 ± 2.48	8.93 ± 2.37	1.000 ^§^	0.02
Immediate post-MS	3.96 ± 1.72	5.11 ± 2.26	0.046 ^§^	0.57	3.96 ± 1.72	5.02 ± 2.12	0.068 ^§^	0.57
1-month post-MS	4.13 ± 2.52	5.00 ± 2.51	0.635 ^§^	0.35	4.16 ± 2.54	5.07 ± 2.58	0.614 ^§^	0.36
HAD-anxiety								
Baseline	9.29 ± 3.02	10.16 ± 2.81	0.162	0.30	9.29 ± 3.02	10.16 ± 2.81	0.162	0.30
Immediate post-MS	5.78 ± 2.52	8.58 ± 2.42	<0.001	1.13	5.78 ± 2.52	8.52 ± 2.43	<0.001	1.11
1-month post-MS	5.80 ± 3.42	8.31 ± 2.49	<0.001	0.84	5.77 ± 3.46	8.40 ± 2.53	<0.001	0.86
HAD-depression								
Baseline	8.76 ± 3.28	8.27 ± 3.22	0.477	0.15	8.76 ± 3.28	8.27 ± 3.22	0.477	0.15
Immediate post-MS	3.89 ± 3.14	5.47 ± 3.96	0.039	0.44	3.89 ± 3.14	5.26 ± 3.99	0.077	0.38
1-month post-MS	4.62 ± 3.59	5.00 ± 3.40	0.610	0.11	4.64 ± 3.64	5.05 ± 3.48	0.594	0.12
PHLMS-total								
Baseline	58.00 ± 6.78	57.56 ± 7.98	0.777	0.06	58.00 ± 6.78	57.56 ± 7.98	0.777	0.06
Immediate post-MS	69.27 ± 7.95	65.49 ± 6.99	0.019	0.50	69.27 ± 7.95	65.33 ± 7.19	0.018	0.52
1-month post-MS	66.73 ± 7.84	65.27 ± 7.82	0.377	0.19	66.70 ± 7.93	65.31 ± 8.07	0.421	0.17
PHLMS-awareness								
Baseline	32.73 ± 6.27	31.51 ± 6.51	0.367	0.19	32.73 ± 6.27	31.51 ± 6.51	0.367	0.19
Immediate post-MS	38.09 ± 4.38	36.60 ± 4.58	0.119	0.33	38.09 ± 4.38	36.57 ± 4.60	0.119	0.34
1-month post-MS	36.69 ± 4.57	36.49 ± 5.43	0.850	0.04	36.77 ± 4.58	36.38 ± 5.60	0.723	0.08
PHLMS-acceptance								
Baseline	25.27 ± 6.57	26.04 ± 6.19	0.565	0.12	25.27 ± 6.57	26.04 ± 6.19	0.565	0.12
Immediate post-MS	31.18 ± 5.92	28.89 ± 5.99	0.072	0.38	31.18 ± 5.92	28.76 ± 5.96	0.061	0.41
1-month post-MS	30.04 ± 7.17	28.78 ± 6.64	0.387	0.18	29.93 ± 7.22	28.93 ± 6.82	0.510	0.14
QOL-total								
Baseline	81.38 ± 10.98	82.98 ± 11.49	0.501	0.14	81.38 ± 10.98	82.98 ± 11.49	0.501	0.14
Immediate post-MS	96.40 ± 8.81	94.00 ± 13.10	0.311	0.21	96.40 ± 8.81	94.57 ± 12.96	0.447	0.17
1-month post-MS	96.58 ± 10.99	93.20 ± 14.41	0.215	0.26	96.50 ± 11.11	93.07 ± 14.82	0.227	0.26
QOL-physical								
Baseline	22.91 ± 3.26	23.67 ± 3.31	0.278	0.23	22.91 ± 3.25	23.67 ± 3.31	0.278	0.23
Immediate post-MS	27.49 ± 2.83	26.31 ± 3.47	0.081	0.37	27.49 ± 2.83	26.52 ± 3.34	0.148	0.31
1-month post-MS	27.31 ± 2.95	26.02 ± 4.00	0.085	0.37	27.27 ± 2.97	26.10 ± 4.09	0.129	0.33
QOL-psychological								
Baseline	17.89 ± 2.80	17.82 ± 2.68	0.908	0.03	17.89 ± 2.80	17.82 ± 2.68	0.908	0.03
Immediate post-MS	21.98 ± 2.81	21.33 ± 3.44	0.333	0.21	21.98 ± 2.81	21.45 ± 3.42	0.434	0.17
1-month post-MS	22.04 ± 2.94	21.22 ± 3.91	0.263	0.24	22.00 ± 2.96	21.14 ± 4.02	0.261	0.24
QOL-social								
Baseline	9.87 ± 1.93	9.87 ± 2.21	>0.999	<0.01	9.87 ± 1.93	9.87 ± 2.21	>0.999	<0.01
Immediate post-MS	11.49 ± 1.58	10.71 ± 2.02	0.045	0.43	11.49 ± 1.58	10.64 ± 2.07	0.036	0.46
1-month post-MS	11.47 ± 1.94	10.84 ± 2.07	0.144	0.31	11.48 ± 1.96	10.79 ± 2.12	0.120	0.34
QOL-environment								
Baseline	24.87 ± 4.95	25.62 ± 4.65	0.458	0.16	24.87 ± 4.95	25.62 ± 4.65	0.458	0.16
Immediate post-MS	28.33 ± 3.30	28.62 ± 4.90	0.744	0.07	28.33 ± 3.30	28.88 ± 4.79	0.540	0.13
1-month post-MS	28.56 ± 4.30	28.24 ± 4.90	0.750	0.07	28.57 ± 4.35	28.19 ± 5.01	0.709	0.08

Abbreviations: CBI, Thai version of the Copenhagen Burnout Inventory; ST-5, the Stress Test Questionnaire; HAD-anxiety, the Thai version of HADS anxiety subscale; HAD-depression, the Thai version of HADS depression subscale; PHLMS, the Thai version of Philadelphia Mindfulness Scale; QOL, the Thai abbreviated version of World Health Organization quality of life (WHOQOL-BREF-THAI). * Comparison of outcomes between Group A at week 4 and Group B at week 12. ** Comparison of outcomes between Group A at week 8 and Group B at week 16. ^a^ at immediate post-MS. ^b^ at one-month post-MS. ^§^ Bonferroni correction was used to adjust *p*-value of primary outcomes (CBI-total and ST-5). Adjusted *p*-value = original *p*-value × 6 (2 outcomes × 3 time points).

**Table 5 healthcare-10-02532-t005:** Program feedback.

	Usefulness of MS			Usefulness of PSA			User-Friendliness of the Platform			Overall Program Satisfaction		
	Group A n = 45	Group B n = 45	All n = 90	Group A n = 45	Group B n = 45	All n = 90	Group A n = 45	Group B n = 45	All n = 90	Group A n = 45	Group B n = 45	All n = 90
Mean score, mean ± SD	4.33 ± 0.60	4.13 ± 0.69	4.23 ± 0.65	4.07 ± 0.69	3.64 ± 0.57	3.86 ± 0.66	4.27 ± 0.65	4.07 ± 0.81	4.17 ± 0.74	4.47 ± 0.63	4.22 ± 0.74	4.34 ± 0.69
Score 5, n (%)	18 (40.0%)	14 (31.1%)	32 (35.6%)	12 (26.7%)	2 (4.4%)	14 (15.6%)	17 (37.8%)	15 (33.3%)	32 (35.6%)	24 (53.3%)	18 (40.0%)	42 (46.7%)
Score 4, n (%)	24 (53.3%)	23 (51.1%)	47 (52.2%)	24 (53.3%)	25 (55.6%)	49 (54.4%)	23 (51.1%)	19 (42.2%)	42 (46.7%)	18 (40.0%)	19 (42.2%)	37 (41.1%)
Score 3, n (%)	3 (6.7%)	8 (17.8%)	11 (12.2%)	9 (20.0%)	18 (40.0%)	27 (30.0%)	5 (11.1%)	10 (22.2%)	15 (16.7%)	3 (6.7%)	8 (17.8%)	11 (12.2%)
Score 2, n (%)	-	-	-	-	-	-	-	1 (2.2%)	1 (1.1%)	-	-	-
Score 1, n (%)	-	-	-	-	-	-	-	-	-	-	-	-

Abbreviations: MS, Mindful Senses; PSA, Psychological Self-Help Articles.

**Table 6 healthcare-10-02532-t006:** Guided mindfulness practice audio listening statistics.

	Group A Median (IQR) n (%)		Group B Median (IQR) n (%)	
	Week 1–4	Week 5–8	Week 9–12	Week 13–16
Total audios listening, minutes	634.9 (292.4, 847.9)	13.6 (0.0, 83.7)	285.8 (126.5, 659.8)	0.0 (0.0, 34.2)
0–199.9	9 (20.0%)	40 (88.9%)	18 (40.0%)	42 (93.3%)
200–399.9	7 (15.6%)	4 (8.9%)	10 (22.2%)	2 (4.4%)
400–599.9	5 (11.1%)	1 (2.2%)	5 (11.1%)	1 (2.2%)
600–799.9	11 (24.4%)	-	4 (8.9%)	-
800–999.9	9 (20.0%)	-	5 (11.1%)	-
1000–1199.9	3 (6.7%)	-	1 (2.2%)	-
1200–1399.9	1 (2.2%)	-	1 (2.2%)	-
1400–1599.9	-	-	1 (2.2%)	-
Total audio listening, times	52.0 (28.0, 72.5)	1.0 (0.0, 9.0)	26.0 (9.5, 54.0)	0.0 (0.0, 3.0)
0–20	9 (20.0%)	40 (88.9%)	20 (44.4%)	43 (95.6%)
21–40	7 (15.6%)	4 (8.9%)	10 (22.2%)	2 (2.2%)
41–60	12 (26.7%)	1 (2.2%)	5 (11.1%)	-
61–80	8 (17.8%)	-	7 (15.6%)	-
81–100	8 (17.8%)	-	2 (4.4%)	-
101–120	1 (2.2%)	-	-	-
121–140	-	-	1 (2.2%)	-
Total days of ≥ 3 audio listening, days	8.0 (2.5, 16.0)	0.0 (0.0, 0.5)	2.0 (0.0, 8.5)	0.0 (0.0, 0.0)
0–4	16 (35.6%)	42 (93.3%)	29 (64.4%)	45 (100%)
5–8	7 (15.6%)	2 (4.4%)	5 (11.1%)	-
9–12	7 (15.6%)	-	1 (2.2%)	-
13–16	5 (11.1%)	1 (2.2%)	5 (11.1%)	-
17–20	3 (6.7%)	-	2 (4.4%)	-
21–24	-	-	-	-
25–28	7 (15.6%)	-	3 (6.7%)	-

## Data Availability

The data presented in this study are available in Appendix A.

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
