# Peer review of "Efficacy and Feasibility of the Minimal Therapist-Guided Four-Week Online Audio-Based Mindfulness Program ‘Mindful Senses’ for Burnout and Stress Reduction in Medical Personnel: A Randomized Controlled Trial"

_healthcare, 2022, doi:10.3390/healthcare10122532_

Round 1

Reviewer 1 Report

This study examined the feasibility and efficacy of a therapist-guided four-week mindfulness-based intervention for burnout and stress in medical professionals. The program produced strong results with minimal drop-out. I applaud the authors for their effort; it is obvious that a lot of work went into this project and manuscript. Overall, the paper is well-written and will make a nice contribution to the literature. Suggestions are below.

·         Minimal information on the intervention is provided in the text. This includes some basic information (E.g., how long were audio recordings? what are “front images”? was BV formally trained in a certain mindfulness-based approach?). Although examples of daily messages are provided in appendixes, it would be helpful to provide more information in text on how the intervention addressed burnout and stress.

·         How much feedback did the provider give to participants? It would be helpful if the authors provided an average and range for number of responses across participants. While the authors refer to their intervention as “minimal therapist-guided,” replying to participants’ inquiries has potential of being time demanding.

·         How was family-wise statistical error controlled?

·         Another limitation is that one author provided the intervention, so generalizability outside of this provider is unknown.

Reviewer 2 Report

I appreciate the authors for doing a meditation intervention specifically for medical professionals, which is considered to be very valuable and meaningful, and this research expand the reference of meditation training for reducing burnout and stress, . However, I think there are some details that could be refined.

Line 17: Given the experimental design later, group B also measured burnout and stress at weeks 12 and 16, which would be more adequately and completely if represented here.

Line 18: Is it appropriate to mention other variables? even if they are not statistically significant, but the abstract should reflects that what you have done in this research, which is generally known when others read this paper (e.g. depression, anxiety, mindfulness and quality of life also improved in your experimental results).

Line 78: The logic of the hypothesis needs to be adjusted, the second hypothesis: PSA + MS will promote the effects of the MS program, but according to your experimental setup, The data of week 4 and week 8 are better in group A. It can only prove that the effects of MS program is great, or that MS program promotes the effects of PSA. It is not sufficient stating that PSA promotes the effects of MS program (if you want to prove that PSA promotes the effects of MS program, there should be one group that attends MS only, while the other attends PSA+MS program, in order to illustrate that PSA could improve MS program).

Line 382: The word “significantly” here would be better represented as “statistically significant”, including other descriptions of “significant” results in the manuscript, so as to not be confused with the common sense of significant.

Line 466: 35% of group A did not complete the required meditation training and so does 62% of group B, this circumstance may have some effect on the results in that group A(MS+PSA) is better than group B(MS after PSA), so maybe attend MS+PSA program is just better than not doing much meditation training?

Line 532: “the MS program strengthened participants resilience toward challenges in life” This inference is not sufficient, because there is no measurement of resilience, or if you could please add more details.

Line 354: I'm sorry I feel your expression here is not very clear. Does it means that group A did not change, while group B INCREASE burnout and back to baseline? I suggest you can express related parts more clear.

Line 434: I suggest to provide more decimal like 5.0 of 5, or such?
